# Alteration of Inflammatory Parameters and Psychological Post-Traumatic Syndrome in Long-COVID Patients

**DOI:** 10.3390/ijerph19127103

**Published:** 2022-06-09

**Authors:** Irma Clemente, Gaia Sinatti, Antonio Cirella, Silvano Junior Santini, Clara Balsano

**Affiliations:** Interdisciplinary BioMedical Group on Artificial Intelligence IBMAI, School of Emergency Medicine, Department MeSVA, University of L’Aquila, 67100 L’Aquila, Italy; irma.clemente@graduate.univaq.it (I.C.); gaia.sinatti@graduate.univaq.it (G.S.); antonio.cirella@graduate.univaq.it (A.C.); silvanojunior.santini@univaq.it (S.J.S.)

**Keywords:** long-COVID, SARS-CoV-2, post-traumatic syndrome, inflammatory parameters, psychopathology, SCL-90, COVID-19, ferritin, sleep disorder

## Abstract

The aim of our study is to evaluate the correlation between the psychological status of patients recovered from SARS-CoV-2 (Severe Acute Respiratory Syndrome Coronavirus 2) infection (long-COVID patients) and their inflammatory status. Three months after hospital discharge, ninety-three patients were recruited and categorized into two distinct populations: control and long-COVID (COrona VIrus Disease) group. Patients belonging to the control group presented with an entering diagnosis of cardiovascular, metabolic, or respiratory disease and a negative history of SARS-CoV-2 infection, whereas the long-COVID population presented with a severe SARS-CoV-2 infection treated in the sub-intensive Care Unit. Psychological evaluation was performed through the administration of the Symptom Checklist-90 (SCL90) and LDH (Lactate dehydrogenase), ferritin, CRPhs (C-high sensitivity Reactive Protein), NLR (Neutrophil-to-lymphocyte ratio), PLR (Platelet-to-lymphocyte ratio), and SII (systemic immune-inflammation index) were investigated. We highlighted that beyond the first three months after contagion, patients recovered from SARS-CoV-2 infection are characterized by the persistence of a systemic inflammatory state and are at high risk for developing somatization, depression, anxiety, and sleep disturbances. Interestingly, ferritin value was strongly correlated with sleep disorders (*p* < 0.05). Our study emphasizes how COVID-19 strategies for risk stratification, prognosis, and therapy management of patients should be implemented with a psychological follow-up.

## 1. Introduction

The spread of the SARS-CoV-2 (Severe Acute Respiratory Syndrome Coronavirus 2) pandemic has affected countless people since December 2019 and still is nowadays a problem that plagues the entire world. Long-COVID (COrona VIrus Disease) or post-covid syndrome, a term coined in May 2020, has been defined by the World Health Organization (WHO) as the set of new, returning, or ongoing health issues people can experience four or more weeks following initial SARS-CoV-2 infection [1,2]. Long-COVID has been described both in severe cases treated in sub-intensive or intensive care units and in cases that did not require hospitalization. It is very worrying that post-covid syndrome has been seen even in asymptomatic patients infected by SARS-CoV-2 [3]. It has been reported that 87% of recovered patients showed persistence of at least one symptom even at 60 days [4,5]. Patients complain of long-term disabling symptoms affecting the musculoskeletal (asthenia, muscle weakness, easy fatigue), respiratory (shortness of breath, persistent cough), cardiovascular (chest pain, tachycardia), and central nervous system (cognitive impairment, memory impairment, Guillain Barre syndrome); these can last for a long time and greatly lengthen the time of a rapid functional recovery [6]. Moreover, a growing recognition of psychopathological symptoms has been reported among patients who recovered from SARS-CoV-2, detecting post-traumatic stress disorder (PTSD), psychosis, anxiety, depression, low mood, insomnia, and obsessive-compulsive symptoms [7]. As is well-known through several studies, coronaviruses such as SARS (Severe Acute Respiratory Syndrome) and MERS (Middle East Respiratory Syndrome) are neurotropic, and “cytokine storms” are involved in neuroinflammation not uniquely in the periphery but also in the central nervous system, causing psychopathological symptoms [8,9]. Here, we explored the psychopathological impact of COVID-19 in survivors, also considering the effect of clinical and inflammatory predictors [10]. Interaction between immunological mechanisms, fear of illness, the uncertainty of the future, stigma, traumatic memories of severe illness, and social isolation experienced by patients during the COVID-19 pandemic are significant psychological stressors that can give rise to a systemic inflammatory state [11]. Today we know for certain, through a significant number of studies [12,13], of the influence of inflammatory biochemical markers in psychopathology. At present, evidence at medium-term follow-up about their correlation to the psychological status of patients who recovered from SARS-CoV-2 infection is still lacking. The aim of our study is to evaluate the inflammatory status of long-COVID patients and their psychological profile.

## 2. Materials and Method

### 2.1. Study Design and Population

Our study is an observational cohort study including 93 patients. We enrolled 48 patients (14 women, 34 men), aged between 40 and 85 years, three months after (95 ± 10 days) their discharge from the COVID sub-intensive Cares Unit of the San Salvatore Hospital of L’Aquila (Local Health Boards 1 of Abruzzo L’Aquila-Avezzano-Sulmona) due to severe SARS-CoV-2 infection [1]. In total, 93% of these patients were dismissed after the resolution of major symptoms and with a negative RT-PCR (Reverse transcriptase-polymerase chain reaction) nasopharyngeal swab; 7% were discharged with the resolution of symptoms but still positive RT-PCR nasopharyngeal swab. Patients were expected to be negative for SARS-CoV-2 RT-PCR to access the study evaluations. The control group consisted of 45 patients (23 women, 22 men), aged between 40 and 80 years, three months (93 ± 9 days) after discharge from the San Salvatore Hospital of L’Aquila (Local Health Boards 1 of Abruzzo L’Aquila-Avezzano-Sulmona). The patients had an entering diagnosis of cardiovascular, metabolic, or respiratory disease, implying a stressful experience and a negative history of SARS-CoV-2 infection. None of the enrolled patients had an entering or following sepsis diagnosis. Underlying diseases have been categorized as comorbidities (Table 1) including: hypertension, diabetes, cardiovascular disease, asthma, chronic obstructive pulmonary disease, obesity, and smoking history. At three months after discharge, all study participants underwent, upon informed consent, the administration of the Symptom Checklist 90 questionnaire (SCL-90) and blood sampling for haemato-chemical tests.

### 2.2. Evaluation of Inflammatory Parameters

We evaluated routine clinical hematology and the following inflammatory parameters: LDH, ferritin, CRPhs, NLR, PLR, and SII. SII index was calculated with the formula SII = (P × N)/L, where P, N, and L refer to peripheral platelet, neutrophil, and lymphocyte counts, respectively. Neutrophil Lymphocyte Ratio (NLR) is represented by the neutrophil/lymphocyte ratio, whereas Platelet Lymphocyte Ratio (PLR) is represented by platelet/lymphocyte ratio.

### 2.3. Psychological Evaluation

Psychological evaluation was performed through the administration of the Symptom Checklist-90 (SCL90), a validated self-report questionnaire: it is a 90-item questionnaire used to help evaluate a broad range of psychological problems and symptoms, and it is a relatively brief psychometric instrument published by the Clinical Assessment division of the Pearson Assessment & Information group. SCL-90 measures 10 primary symptom dimensions and is designed to provide an overview of a patient’s symptoms and their intensity at a specific point in time. The ten subscales are: somatization, obsessive compulsion, interpersonal sensitivity, depression, anxiety, hostility, phobic anxiety, paranoid ideation, psychoticism, and sleep disorders. The test is a 90-item questionnaire; each item is scored on a scale from 0 to 4 (Likert scale) based on how much an individual was bothered by each of them in the last week: 0 = Not at all; 1 = A little bit; 2 = Moderately; 3 = Quite a bit; 4 = Extremely. “Obsession” investigates intrusive thoughts and compulsive actions, “sensibility” feelings of inadequacy and inferiority in relationship with others, “hostility” stands for aggressiveness towards others, “phobic anxiety” for fears related to specific stimuli, “paranoid ideation” for projections to others and persecutory cognitions, “psychoticism” is to be understood as psychotic and schizophrenic behaviors, “anxiety” for anxiety symptoms and experienced tensions, “somatization” for distress related to one’s body/physiological experiences, “depression” stands for the broad spectrum of symptoms concomitant with a depressive syndrome, and “sleep disorder” investigates insomnia, disturbed sleep, and early awakening.

### 2.4. Statistics

Clinical and demographic data were expressed as means ± standard deviations. Categorical variables were reported as a percentage and analyzed by the chi-square. As a measure of the association between inflammatory status and psychological disorders, we carried out Pearson’s correlation coefficient; in the case of more than one independent variable, multiple logistic regression was performed. *p*-values < 0.05 were considered statistically significant. Microsoft Excel 365, StatSoft Statistica 10, and GraphPad Prism 6 packages were used for data processing, statistical analyses, and visualization.

## 3. Results

A description of the demographic, hematological, and inflammatory parameters of our cohort of 93 patients is reported in Table 1. Of the whole cohort of long-COVID patients, 70.83% were male (vs. 48.88% control group). The age and BMI (Body Mass Index) were similar between the two genders in the long-COVID group (mean age 62.90 ± 9.22 and BMI mean 28.46 ± 4.50 vs. mean age 62.31 ± 18.78 and BMI mean 26.32 ± 2.68). After three months discharge, among all the inflammatory parameters analyzed, LDH and ferritin displayed statistically significant higher values with respect to the control group (*p* < 0.001 and *p* < 0.01, respectively). Among the inflammatory indices, PLR, NLR, and systemic immune inflammation (SII), we noticed that SII was significantly higher in the long-COVID group with respect to the control group (*p* = 0.007). Furthermore, ferritin correlates with sleep disorders (*p* < 0.05) in the long-COVID group. No effect (*p* = 0.2), implying sex as a potential influencing factor, was found when we performed multiple logistic regression to evaluate the relationship between ferritin levels and sleep disturbance. No other significant correlations were found between the inflammatory indexes and the psychopathological items. Comorbidity overall score was similar between the control and the long-COVID group (75.55% vs. 79.16%), showing no influence on our results. The prevalence of hypertension, diabetes, cardiovascular disease, asthma, chronic obstructive pulmonary disease, obesity, smoking history in long-COVID population were similar respect to the control group: 47.9% vs. 51.1%, 10.41% vs. 8.88%, 18.75% vs. 13.33%, 4.16% vs. 4.44%, 2.08% vs. 2.22%, 79.16% vs. 75.55%, 16.66% vs. 22.22%, respectively. Comorbidities showed no influence on our results. In Table 2, we show significant differences in the distribution of psychopathological features in the two groups of patients: 66.67% of the long-COVID group scored positively in at least one dimension (vs. 53.3% in the control group), 54.16% in two (vs. 33.32% in the control group), 35.42% in three (vs. 22.23% in the control group), and 29.17% in four (vs. 13.34% in the control group). Accordingly, the prevalence of anxiety, somatization symptoms, and sleep disorders in post-COVID population were higher respect to the control group: 25.00% vs. 9.09% (*p* < 0.05), 47.92% vs. 27.27% (*p* < 0.05) and 66.67% vs. 38.64% (*p* < 0.01), respectively. Psychopathological differences between males and females were highlighted in the post-COVID population: percentage of somatization was present in 64.28% of women vs. 41.47% of men (*p* < 0.05); anxiety 50% of women vs. 14.70 % of men; depression in 50% of women vs. 20.58% of man; finally, sleep disorders were found in 85.71% of women vs. 58.82% of men (Table 2). In Figure 1, we highlight the significant differences in the score of anxiety, somatization, depression, and sleep disorders in the two studied groups.

## 4. Discussion

Our work is in line with the global efforts to develop better diagnostics and prognostics for SARS-CoV-2 infection ranging from monitoring susceptibility, progression and resolution of disease, health condition, and outcomes. Considering the alarming impact of COVID-19 on mental health, the current insight of inflammation in psychological sequelae and the present observation of worse inflammation leading to worse depression is a subject worthy of attention. Our study has correlated the impact of psychopathology with some inflammatory parameters at three months items follow-up after sub-intensive hospitalization of SARS-CoV-2 infected patients. COVID-19 survivors presented a high prevalence of emergent psychopathological sequelae, with 66.67% of the sample presenting a pathological score for at least one disorder on the SCL90 test. The prevalence of depression, anxiety, mood disorders, and sleep disorders has been proved even in previous studies [14,15]. We highlighted that beyond the first three months after contagion, individuals who recovered from SARS-CoV-2 infection are characterized by the persistence of a systemic inflammatory state and are at high risk for developing somatization, depression, anxiety, and sleep disturbances. In particular, we found out that the median baseline serum LDH and ferritin levels were significantly higher three months after SARS-CoV-2 infection [16]. Interestingly enough, ferritin value was strongly correlated with sleep disorders, independently of gender. Long-COVID group at the SCL-90 analysis has higher levels of anxiety, depression, somatization symptoms as well as sleep disorders (*p* < 0.05, *p* < 0.05, *p* < 0.05 and *p* < 0.01, respectively). The correlations between the inflammatory status and psychological disorders are in line with the study performed by Kim KM et al. in old Korean patients in whom high ferritin levels were associated with sleep disturbances, stress, depression, and suicidal ideation [17,18]. Women are characterized by higher score values for most psychological disorders evaluated by the SCL-90 test; differences are mainly evident for anxiety and sleep disorders (see Table 2). As has been pointed out in several studies, sex differences could arise because males could be more hesitant to report emotional symptoms due to socio-cultural pressures or expectations [19,20]. It is important to note that although the values of inflammatory parameters are higher in males, psychological disorders are more evident in females. One explanation might be that men usually show less psychological discomfort than women; in fact, women tend to have higher rates of assuming internalizing disorders (depression, anxiety, and mood swings), while men experience is characterized by more externalizing symptoms (headache, fatigue, backpain, etc.) [21].

## 5. Conclusions

Our work underlines that care pathways for long-COVID patients should be implemented with a psychopathologic follow-up. The demographic composition of our cohort was characterized by a higher number of males, reflecting the prevalence of SARS-CoV-2 infection [22]; despite this, we found out that sex does not influence the correlation between ferritin levels and sleep disturbances. Our work is in line with the literature; survivors of SARS-CoV-2 are at risk of psychiatric sequelae [23]. Furthermore, early SARS-CoV-2 survivor patients have been evaluated by SCL90, which proved to be useful in underlining the importance of early mental health and psychosocial services at the stage of inpatient treatment [24]. Due to the reliability and inexpensive nature of the SCL-90 test, it could help in planning better post-acute COVID-19 strategies for risk stratification, prognosis, and therapy management of patients.

## 6. Study Limitations

This study has some limitations. No individual psychological test can be sufficiently reliable, so there is a need to improve knowledge on psychosomatics tests useful for accurately monitoring a patient’s post-COVID wellbeing.

## Figures and Tables

**Figure 1 ijerph-19-07103-f001:**
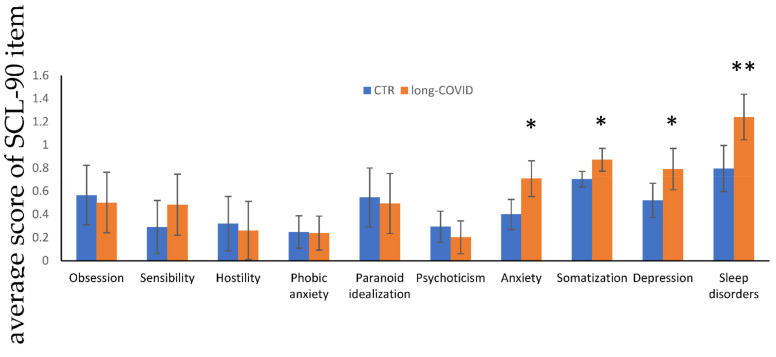
Psychological parameters analyzed by SCL-90 test. Values are expressed as means ± SD and data were analyzed by a *t*-test analysis (* *p* < 0.05; ** *p* < 0.01).

**Table 1 ijerph-19-07103-t001:** Values are reported as mean values for the continuous variables and percentages for the categorical variables. Statistical significance was assessed by *p* value (*p*) thresholds: ** *p* < 0.01; *** *p* < 0.001 (*t*-test analysis).

Characteristics	Control Group			Long-COVID		
	Overall (n = 45)	Male (n = 22)	Female (n = 23)	Overall (n = 48)	Male (n = 34)	Female (n = 14)
Age (years)	60.62 ± 17.37	62.31 ± 18.78	59.50 ± 16.35	62.54 ± 10.18	62.90 ± 9.22	61.69 ± 12.57
BMI	26.04 ± 3.43	26.32 ± 2.68	25.77 ± 4.05	27.91 ± 4.26	28.46 ± 4.05	27.14 ± 4.02
LDH (UI/L)	189.33 ± 42.36	192.00 ± 38.99	187.00 ± 47.08	238.01 ± 60.01 ***	241.00 ± 64.51	223.50 ± 36.29
Ferritin (ng/mL)	159.27 ± 169.17	216.11 ± 175.04	110.34 ± 153.05	298.09 ± 209.02 **	349.99 ± 208.82	135.91 ± 102.61
CRP (mg/dL)	0.55 ± 0.61	0.62 ± 0.76	0.51 ± 0.46	0.73 ± 0.75	0.75 ± 0.79	0.66 ± 0.67
SII	477.72 ± 260.63	441.22 ± 313.36	512.63 ± 198.80	715.91 ± 267.98 **	757.45 ± 742.08	683.73 ± 261.09
PLR	205.78 ± 152.31	195.29 ± 149.00	226.56 ± 146.40	186.07 ± 75.56	193.51 ± 83.09	162.39 ± 37.50
NLR	2.04 ± 1.35	2.16 ± 1.66	2.23 ± 1.33	2.66 ± 1.41	2.71 ± 1.53	2.49 ± 0.88
Leucocytes (×10^3^/μL)	7.50 ± 1.84	8.06 ± 1.92	6.99 ± 1.64	6.17 ± 1.55	6.12 ± 1.58	6.32 ± 1.50
Hemoglobin (g/dL)	14.27 ± 1.58	14.82 ± 1.72	13.75 ± 1.25	14.83 ± 4.20	14.40 ± 1.35	16.14 ± 8.31
Platelets (×10^3^/mcL)	248.64 ± 68.68	247.72 ± 69.98	249.52 ± 68.98	258.37 ± 74.23	260.74 ± 77.17	250.82 ± 66.85
Lymphocytes (×10^3^/μL)	1.57 ± 0.80	1.66 ± 0.76	1.48 ± 0.84	1.46 ± 0.34	1.43 ± 0.36	1.56 ± 0.27
Neutrophils (×10^3^/μL)	2.86 ± 1.52	2.93 ± 1.70	2.79 ± 1.35	3.61 ± 1.31	3.66 ± 1.41	3.46 ± 1.00
Glycemia (mg/dL)	85.59 ± 13.85	100.55 ± 16.79	90.64 ± 8.18	93.11 ± 15.42	93.09 ± 16.15	93.17 ± 11.96
ALT (UI/L)	22.38 ± 15.62	21.80 ± 8.75	23.57 ± 25.85	30.19 ± 25.83	32.85 ± 28.05	20.44 ± 11.63
AST (UI/L)	22.83 ± 20.21	24.90 ± 21.05	20.57 ± 10.80	27.93 ± 17.71	30.78 ± 18.64	17.78 ± 8.61
Creatinin (mg/dL)	0.83 ± 0.20	0.95 ± 0.22	0.72 ± 0.07	0.88 ± 0.20	0.91 ± 0.20	0.75 ± 0.16
Comorbidities	75.55%	81.81%	69.56%	79.16%	81.57%	68.33%

**Table 2 ijerph-19-07103-t002:** Values are reported as percentage for the categorical variables. Statistical significance was assessed by *p* value (*p*) thresholds: * *p* < 0.05; ** *p* < 0.01 (*t*-test analysis).

Characteristics	Control Group			Long-COVID		
	Overall (n = 45)	Male (n = 22)	Female (n = 23)	Overall (n = 48)	Male (n = 34)	Female (n = 14)
Obsession	15.90%	4.54%	27.27%	14.58%	11.76%	21.42%
Sensibility	2.27%	4.54%	0%	22.91%	14.70%	42.85%
Hostility	9.09%	9.09%	9.09%	8.33%	5.88%	14.28%
Phobic anxiety	2.27%	0%	4.54%	8.33%	5.88%	14.28%
Paranoid ideation	13.63%	18.18%	9.09%	8.33%	5.88%	14.28%
Psychoticism	6.81%	9.09%	4.54%	4.16%	2.94%	7.14%
Anxiety	9.09%	4.54%	13.63%	25% *	14.70%	50% *
Somatization	27.27%	13.63%	40.90%	47.91% *	41.17% *	64.28%
Depression	9.09%	4.54%	13.63%	29.16% *	20.58% *	50% *
Sleep disorders	38.63%	40.90%	36.36%	66.66% **	58.82%	85.71% **

## Data Availability

The datasets used and/or analyzed during the current study are available from the corresponding author on reasonable request.

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
