# Peer review of "Alteration of Inflammatory Parameters and Psychological Post-Traumatic Syndrome in Long-COVID Patients"

_ijerph, 2022, doi:10.3390/ijerph19127103_

Round 1

Reviewer 1 Report

The aim of the presented study is to evaluate the inflammatory status of long-COVID patients and their psychological profile. However simply description design without study on the relationship between variables prominently weakens the value of the study. There are several major limitations to explore the relationship between inflammatory status and the psychological profile.
1.The authors did not explain the statistical methods to explore the relationship between inflammatory status and the psychological profile. Instead of showing the results of the relationship above, the authors only said that "ferritin correlates with sleep disorders (p<0.05) in long-COVID group."
2. The authors did not deal with the effects of confounding factors. For examples, the sex distribution was statistically different bwteen long-COVID group and control group, which might have effects on the ferritin levels and psychological symptoms. It is very difficult to clarify that the difference of the psychological symptoms between the long-COVID group and control group was contributed by COVID infection or confoundings, such sex and underlying diseases.
3.Besides, some important references did not collect and discuss. Such as (1).Brain Behav Immun. 2021 Oct;97:328-348. doi: 10.1016/j.bbi.2021.07.018. Epub 2021 Jul 30. (2)Transl Psychiatry. 2021 May 17;11(1):290. doi: 10.1038/s41398-021-01416-5.

Author Response

We sincerely thank Reviewer 1 for her/his insightful comment. We really appreciate how her/his comments improved our paper. We hope the changes and explanations will meet your requirements.

We have addressed all the comments as explained below:

1.We added the statistical methods to explore the relationship between inflammatory status and the psychological profile (pag.3, line 113-118)

  1. We agree and we know that there is a certain sex difference between males and females in both the analyzed groups; this is why we tried to choose two homogenous groups between long-COVID and control patients. We make this point on pag. 6 lines 215-217.
  2. We found very interesting the notables works,so we added the suggested references in the manuscript. Look at discussion section at pag. 6, line 207 and line 208 with the corresponding references.

Reviewer 2 Report

This study by Irma et al., assesses the risk of developing psychological and inflammatory problems in long term Covid exposed patients. I found this study important for the management of post covid exposed patients. 

There are minor concerns needs to be answered before this paper should be considered for publication:

1. In the table 1. the values of Leucocytes in overall long-Covid  is 61.17, please check if it is 6.17 or this is written correct?

2. In the Figure 1. Y axis label is missing 

Author Response

We would like to sincerely thank reviewer 2 for her/his time and valuable advice. We really appreciate the improvement her/his comments made on our paper. We made the suggested changes. In particular, we corrected the typing error, the proper value is 6.17 mg/dL in table 1, and the missing Y axis in table 1.

Reviewer 3 Report

I have read this article. Very interesting results. It has been well known that inflammation is associated with psychological disorder. I have only minor comments.

What was the difference beween sepsis and non-sepsis? How many patients meet the criteria of sepsis III? SOFA score should be added for the patient's background.

Why did you pick up SCL90? Are there any recommendation to use this? Of course, it is very difficult the best score for psychological evaluation as mentioned in research limitation section. Please check this review (PMID: 34501316  DOI: 10.3390/jcm10173870). Thre are several recommended psychological evaluation scores. How this SCL 90 exceeds these scores?

Author Response

Dear Reviewer 3 thank you, we really appreciate your comments

Here the explanation you asked:

1.None of our patients from the long-COVID (treated in sub-intensive care unit) or from the control group had an entering or following sepsis diagnosis. We agree with your insightful comment, thus we inserted a specific sentence in this section to clarify the studied population (pag.2, line 77).

  1. SCL90 is a very easily understanding tool for patients, it is a self-reported test and evaluates the severity of both internalizing and externalizing symptoms of mental distress. It gives the possibility of repeated administration. So, it’s useful to measure patients changes and document and evaluate treatment outcomes (psychotherapy and pharmacotherapy) and response trends.

Finally, we used this tool because our studies are going on, and we are able with this method to have an easy follow-up over time.

Thank you for this insightful comment we hope these changes and explanations will meet your requirements. Kindest regards.

Reviewer 4 Report

Irma and colleagues administered the SCL-90 test to individual with long-COVID and individuals recovering from other illnesses.  They also examined numerous lab reports for both groups.  They found that anxiety, somatization, depression, and sleep disorders occurred at a significantly higher frequency among long-COVID patients.  They also found statistically significant differences in LDH, ferritin, and SII lab values.  This is a brief but interesting report that could be substantially and quickly improved by additional analysis of the existing data set and expansion of the results and discussion sections. 

Major Comments:

1.       The manuscript should include some description of the characteristics.  What constitutes “obsession” or “sensibility”?  This paper is at the intersection of infectious disease, immunology, and behavioural sciences.  Authors should provide sufficient background information for readers from a variety of disciplines.

2.       Statistical significance of results for LDH, Ferritin, SII should be more fully explored.  Is there any relationship between these lab tests and the significantly different anxiety, somatization, depression, and sleep disorder scores reported for the long-COVID group?  In the discussion, the authors comment that “ferritin value was strongly correlated with sleep disorders (P<0.05)” (lines 155-156).  This statement is a result and should be in the results section, not the discussion.  Results should be expanded to include complete analysis of possible relationship, and the discussion should be revised to contextualize the results.

3.       The data set does not include a healthy control group of any sort, e.g. no history of recent illness or full recovery from SARS-CoV-2 infection.  If a healthy control group of some sort could be added, I think that it would strengthen the paper.  However, the data set in the paper as it exists now is interesting and moves the field forward.

Minor Comments:

1.       I know an abbreviation list is included at the end, but I think it would be helpful if abbreviations were also be spelled out on first use.  I leave this to the authors’ discretion. 

2.       Lines 46-47 – “neurotrophic” should be spelled “neurotropic”

3.       Line 47 – “cytokines storm” should be “cytokine storm”

4.       Table 2 – I think that “paranoid idealization” should be “paranoid ideation”

5.       The order of categories along the X-axis in Fig. 1 should match the order of categories in Table 2.

Author Response

We would like Reviewer 4 for her/his time and valuable advice. We really appreciate the improvement your comments made on our paper, and we enjoy the idea of working on additional analysis to identify new attractive research fields.

We agree in making the suggested changes:

  1. We have rewritten and expanded the 2.3 section: “Psychological evaluation”

to clarify the description of the characteristics analyzed in SCL90. As you suggested this is needed because the paper is addressed to readers from a variety of disciplines. You can find the description at pag.3, lines 101-109.

  1. Unfortunately, LDH, SII and other inflammatory index, except for ferritin, did not find significant correlation with any of the psychological symptom investigated by SCL90. We clarified this statement at pag. 3, line 135-136.
  2. We moved the statical significance of ferritin and sleep disorders correlation from the discussion section to the result section (see pag. 3., line 130-132).
  3. We agree with the reviewer, our research would be improved by the addition of a control group.

About minor comments:

We defined the abbreviation the first time they appear in the abstract and introduction, you can find them highlighted in yellow. We fixed the several typos and grammatical errors suggested and proofread the paper to eliminate all such errors.

We changed the Fig.1 adding the Y axis and changing the order of the categories according to Table 2 order.

Thank you for this insightful comment we hope these changes and explanations will meet your requirements. Kindest regards.

Round 2

Reviewer 1 Report

Thank the authors for their responses. However, the explanation did not deal with my worries.

When the sex distribution is different between the long-COVID and control groups and sex influences on the ferritin levels, the sex would be considered having a confounding effect on the relationship between ferritin level and long-COVID. It was usually suggested to use regression modeling or stratification analysis to deal with the possible confounding effects. In the response, the authors listed sex difference as limitation but still did not deal with the important confounding effect, which might influence on the study conclusion

Another important confounding factor is the underlying diseases in the two groups but the authors did not mention. I do not know if the difference of inflammatory parameters and psychological symptoms between the long-COVID and control groups contributed from the status of long-COVID or the confounding factors.

The results of the Symptom Checklist-90 (SCL90) should be considered as subjective symptoms but not a diagnosis nor a disorder. And there is a similar question that the authors did not deal with the confounding effects (age and underlying diseases) on the relationship between long-COVID and SCL90 findings.

It is a pity that the conclusion was weaken by the un-resolved confounding effect in this cross-sectional study.

I can not find table 3 in the manuscript.

Author Response

We would like to sincerely thank reviewer 1 for her/his time and valuable advice. We really appreciate the improvement her/his comments made on our paper.We have addressed all the comments as explained below:

1.We performed the multiple logistic regression analysis in case of more than one independent variable, to avoid the confounding effect related to sex (pag.3, lines 118-119; pag.3,lines 134-136; pag.6, lines 201-202; pag.6, lines 219-222).

  1. We added the overall comorbidities percentage for both groups in Table 1 (page 4). We discussed them in “Study design and population” (pag.2, lines 78-80) and in the “Results” section (pag.3, lines 137-143).
  2. We used this tool because our studies are going on, and we are able with this method to have an easy follow-up over time. We would like to underline that we decided to use Symptom Checklist-90 (SCL90) because it is a very easily understanding tool for patients. It is a self-reported test and evaluates both internalizing and externalizing symptoms of mental distress. Moreover, it gives the possibility of repeated administration. So, it’s useful to measure patients changes and document and evaluate treatment outcomes (psychotherapy and pharmacotherapy) and response trends. Finally, as you can see above, we performed a  multiple regression analysis evaluating sex as a possible confounding parameter. Regarding age, the two populations have similar distribution: this should avoid misleading interpretations, as we pointed out at page 3, lines 126-128.
  1. We proofread our work and corrected the drafting error at page 6, line 209.

Kindest regards.

Reviewer 4 Report

The authors have addressed most of my concerns.  The content in the revised manuscript has better explanations for key points which will help readers from different backgrounds.  Lack of a healthy control group continues to be a weakness should not prevent publication of the manuscript.

Minor Comments:

Throughout the manuscript:  "Sars-CoV2" and "SARS-CoV2" should appear as "SARS-CoV-2."  This can be corrected in proof.

Author Response

We sincerely thank Reviewer 4 for her/hisinsightful comment.We proofread our work and corrected all the disease definitions into SARS-CoV-2. Thank you for your time. Kindest regards